# PARDIFF: BRIDGING AUTOREGRESSIVE AND DIFFUSION MODELS FOR ORDER-AGNOSTIC GRAPH GENERATION

## ABSTRACT

Graph generation has long struggled with the trade-off between structural fidelity and permutation robustness: autoregressive models excel in expressivity but break under node-order sensitivity, while diffusion models offer invariance at the cost of directional coherence. We introduce PARDIFF, a Progressive AutoRegressive DIFFusion framework that unifies these strengths through block-wise, order-agnostic generation guided by learned structural decomposition. Unlike prior heuristics, PARDIFF jointly predicts block sizes, ranks nodes, and applies an equivariant diffusion process to each block, aligning AR directionality with diffusion robustness. This reframes graph synthesis as probabilistic reasoning over learned topological partitions, enabling scalable, semantically faithful, and order-agnostic generation across molecular and non-molecular domains without auxiliary features. Experiments show state-of-the-art results on diverse benchmarks, while its modular, latency-aware design supports real-time applications like drug–drug interaction analysis, positioning PARDIFF as a paradigm shift in structured generative modeling. Code is available at: `https://github.com/llmresearch678/Pardiff_M_1`.

## 1 INTRODUCTION

Graphs lie at the heart of modeling complex relational structures across diverse domains—including social networks, biochemical systems, recommendation engines, and cyber-physical infrastructures (Kong et al. (2023); Cohen-Karlik et al.; Li et al. (2024); Chen et al. (2023)). As machine learning advances toward general-purpose, foundation-level models, graph generative modeling has emerged as a pivotal capability—fueling applications in molecular synthesis, protein engineering, and synthetic network design (Niu et al. (2020); Liao et al. (2019a)). Unlike grid-based modalities such as images or text, graphs are inherently combinatorial, permutation-invariant, non-Euclidean, and variable in size. This introduces profound challenges in maintaining structural validity, generalization across topologies, and permutation-consistent generation (Dai et al. (2020); Guo & Zhao (2022)).

To tackle graph generation challenges, prior works span AR models (You et al. (2018); Liao et al. (2019a); Jin et al. (2018)), VAEs, GANs (Roy & Dasgupta (2023; 2024b;a)), and diffusion methods (Du et al. (2021); Jo et al. (2022b); Huang et al. (2022)). AR models excel in controllability but suffer from permutation bias and factorial inference costs (O'Bray et al. (2021); Luo et al. (2021); Honda et al. (2019)). Diffusion models like EDP-GNN (Vignac et al. (2022b)) and GDSS (You et al. (2023)) offer order-agnostic generation via SDEs, yet struggle with discrete structures. DI-GRESS (Vignac et al. (2023)) adopts discrete-state transitions but relies on handcrafted priors. Hybrid methods such as GRAPHARM (Zhang et al. (2021)) partially bridge the gap but impose rigid orderings. No prior approach fully unifies scalability, permutation-invariance, and structural expressivity in a single, efficient framework.

We introduce PARDIFF, a Progressive AR-Diffusion framework that bridges the structural control of autoregression with the robustness of discrete diffusion. Unlike prior models that treat graphs as monolithic, PARDIFF generates them block-wise through dynamically learned topological decompositions—predicting block size and order, then modeling each block with a shared equivariant diffusion process. This aligns generation with natural partial orderings while ensuring seman-

tic fidelity and scalability. To overcome equivariant models' symmetry limitations, we design a noise-guided transition mechanism—akin to simulated annealing—that drives asymmetry formation through structured perturbations, yielding richer and more diverse graphs. Finally, we introduce a higher-order graph transformer with GPT-style parallel training, fusing edge-level reasoning from Provably Powerful Graph Networks with transformer expressivity. Together, these innovations establish PARDIFF as a paradigm shift in graph generation, delivering state-of-the-art results on large-scale benchmarks without handcrafted features or auxiliary supervision.

## 2 PARDIFF: STRUCTURED DIFFUSION FOR PERMUTATION-INVARIANT GRAPH GENERATION

Diffusion-based generative models (Haefeli et al. (2022); Madhawa et al. (2019)) work by gradually adding noise to data until it is completely unrecognizable, and then training a model to reverse this process and recover the original input. Originally developed for continuous data like images, researchers have recently adapted these models to handle discrete data, including graphs (Song et al. (2020); Simonovsky & Komodakis (2018))—structures made of nodes and edges. In the graph context, the process begins with a clean graph $G_0 = \{\mathcal{V}_0, E_0\}$, where $\mathcal{V}_0$ denotes the node features (one-hot vectors encoding categorical attributes such as type or label) and $E_0$ denotes edge features (one-hot encodings of relation types, connection categories, or an explicit "no-edge" token). The no-edge token ensures that graph diffusion models can explicitly represent the absence of a connection between two nodes, making the edge feature space complete. It is critical both for stable noise injection/denoising during training and for generating valid, sparse, and realistic graphs at inference. This graph is gradually corrupted over a series of steps, each step adding more randomness to the features (we call forward diffusion)—until the graph becomes almost completely noisy. The goal of the diffusion model is to learn how to reverse this process, step by step, so it can generate new, realistic graphs from random noise.

The forward diffusion trajectory is described by a sequence of latent variables $G_1, G_2 \cdots G_T$ over $T$ time steps, where $G_t = \{\mathcal{V}_t, E_t\}$ represents the noisy version of $G_0$ at time step $t$. This forward process is modeled by the following Markov chain: $q\big(G_t \mid G_{t-1}\big) = \prod_i q\big(f_t^i \mid f_{t-1}^i\big) \prod_{i,j} q\big(r_t^{ij} \mid r_{t-1}^{ij}\big)$, where $f_t^i$ and $r_t^{ij}$ denote the categorical states of node $i$ and edge $(i,j)$ at time $t$ respectively. $f_t^i, r_t^{ij} \in \{1 \cdots n\}$ with $n$ is the number of states. The learning problem then reduces to parameterizing a reverse-time process (we call it denoising) $p_\phi\big(G_{t-1}|G_t\big)$ that approximates $q\big(G_t|G_{t-1}\big)$ but runs backwards, from unstructured noise $G_t$ to structured samples resembling the original data distribution. In practice, this requires training a denoising network (score function or conditional transition model) that iteratively refines noisy graphs, balancing local consistency (node/edge attributes) and global topology (graph structure).

In this work, we model the reverse (denoising) process using a parameterized transformer neural network with parameters $\phi$ and estimates the backward transition as follows: $p_\phi\big(G_{t-1}|G_t\big) = \prod_i p_\phi\big(f_{t-1}^i|G_t\big) \prod_{i,j} p_\phi\big(r_{t-1}^{ij}|G_t\big)$. Subsequent loss function should balance two things: ($1$) It tries to minimize the difference between the real data and what it generates (via a cross-entropy loss), and ($2$) It also tries to make the learned denoising steps as close as possible to the true underlying reverse steps (via a KL divergence term $\mathcal{D}\big(\cdot \| \cdot\big)$). Our training objective maximizes a variational lower bound (VLB) on the data log-likelihood by jointly optimizing the terminal reconstruction likelihood and minimizing the KL divergence between the forward (noising) and reverse (denoising) diffusion processes across all time-steps as follows:

$$\log p_\phi\big(G_0\big) \geq \mathbb{E}_q\left[\log p_\phi(G_0 \mid G_1)\right] - \mathcal{D}\big[q(G_T \mid G_0) \,\|\, p_\phi(G_T)\big] -$$
$$\sum_{t=2}^{T} \mathbb{E}_q\left[\mathcal{D}\left(q(G_{t-1} \mid G_t) \,\|\, p_\phi(G_{t-1} \mid G_t)\right)\right], \tag{1}$$

where $p_\phi\big(G_T\big)$ is typically set as a fixed uniform noise distribution. Unlike traditional diffusion models that estimate each $p_\phi(G_{t-1} \mid G_t)$ independently, we directly learn $p_\phi\big(G_0|G_t\big)$ and derive all intermediate steps from it. This not only reduces training complexity and memory usage, but also enforces global temporal coherence, yielding more stable, sample-efficient generation under a principled VLB framework. As shown in APPENDIX, this follows from the variational objective.

which enables us to use a cross-entropy (CE) loss at each timestep:

$$\mathcal{L}_{\text{CE},t}(\cdot) = -\mathbb{E}_q\big[\sum_i \log p_\phi\big(f_0^i|G_t\big) + \sum_{i,j} \log p_\phi\big(r_0^{ij}|G_t\big)\big], \tag{2}$$

which we combine with the VLB loss to create a hybrid objective: $\mathcal{L}_t(\cdot) = \mathcal{D}\big(\cdot \| \cdot\big) + \lambda \cdot \mathcal{L}_{\text{CE},t}(\cdot)$, with $\lambda = 0.1$. During generation, a synthetic graph is sampled from $p_\phi(G_T)$ and iteratively denoised via the learned reverse process $p_\phi(G_{t-1} \mid G_t)$ for $t = T$ down to $0$. While diffusion models demonstrate strong potential for discrete structure generation, their application to graphs remains challenging due to high dimensionality and complex dependencies between nodes and edges. Prior works like DIGRESS (Vignac et al. (2023)) address this by incorporating auxiliary structural cues (*e.g.*, spectral eigenvectors, cycle indicators), but these add computational overhead and introduce reliance on domain-specific priors. Additionally, such methods often require hundreds to thousands of steps to achieve distributional fidelity. In contrast, we adopt a simplified discrete-time diffusion approach, which improves memory efficiency and enables exact computation of the variational loss. The complete derivation of the forward and reverse distributions used in our model—$q(G_t \mid G_0)$, $q(G_{t-1} \mid G_t, G_0)$, and $p_\phi(G_{t-1} \mid G_t)$—is provided in APPENDIX.

## 2.1 STRUCTURE-AWARE SEQUENTIAL GRAPH GENERATION

AR models generate graphs step-by-step by breaking down the joint probability into a sequence of conditional decisions—each choice depending on what has already been generated. This approach works well for data with natural order, like text or images. However, graphs are permutation-invariant, meaning their structure does not depend on the order of the nodes. This creates a fundamental mismatch: AR models are sensitive to order, while graphs are not. Early graph generation models like GRAPHRNN (You et al. (2018)) and GRAN (Liao et al. (2019b)) handled this by assigning an artificial node ordering—using methods like breadth-first search, depth-first search, or $k$-core decompositions—to serialize the graph. While these heuristics allow training, they introduce biases that do not reflect the true nature of graph distributions. These approaches often perform well on small or synthetic graphs with regular structures, but struggle to generalize to larger or more complex graphs where order invariance is crucial for accurate modeling.

There are two common strategies to address this: ($1$) Marginalize over all possible node orderings $p(G, \pi)$, but this becomes computationally infeasible because the number of orderings grows factorially. ($2$) Use a fixed, canonical ordering for each graph, but finding such an ordering is as hard as solving the graph isomorphism problem, which is computationally challenging and often dataset-specific. To avoid these limitations, we propose a more flexible and general approach: instead of enforcing a strict global order, we leverage partial structural ordering. The key insight is that not all nodes are equal—some play similar roles based on how they are connected. We group nodes into blocks based on their structural roles, assigning each node a rank or block index via a function $\psi : \mathcal{V} \to \{1, \cdots B\}$, where $B$ is the number of blocks.

During generation, we treat nodes in the same block as structurally interchangeable and generate the graph block by block, not node by node. To maintain coherence and realism, we ensure that each new block connects to the previously generated part of the graph. Formally, we require that the subgraph $G' = \{\mathcal{V}', E'\}; \mathcal{V}' \subseteq \mathcal{V}$ induced by all nodes up to block $b$ is connected: $\forall b \in \{1, \cdots B\}$, $G'\big[\psi^{-1}\big(\le b\big)\big]$ is connected. This approach aligns with how real-world graphs grow—by expanding around existing structures—and avoids the rigidity and bias of fixed orderings. It brings together structural awareness, flexibility, and scalability, offering a more natural and powerful foundation for graph generation.

**Weighted Degree Hashing for Ranking.** To reduce rank collisions and capture broader structural context, we introduce a weighted degree function over $K$-hop neighborhoods. Let $\delta_k(V); V \in \mathcal{V}$ be the number of nodes reachable from node $v$ within exact $K$ hops. Then we define the weighted structural score: $w_K(V) = \sum_{k=1}^{K} \delta_k(V) \cdot |\mathcal{V}|^{K-k}$. This encoding gives greater importance to lower-hop connectivity. Having defined $w_K(V)$, we introduce structural partial order in Algo. 1.

**Theorem 1.** *The structural ranking function $\psi$ (Algo. 1) is permutation-consistent, i.e., for any $G = \{\mathcal{V}, E\}$ and permutation $\pi$ that reorders the nodes of $G$, the ranking satisfies: $\psi\big(\pi \star G\big) = \pi \star \psi(G)$, where $\star$ is the natural action of $\pi$ on both the graph structure and node ranking map.*

**Proof of Theorem 1 is in APPENDIX.** The ranking $\psi(u)$ of a node $u \in \mathcal{V}$ is determined in Algo. 1 from the multi-hop structural weight $w_K(u)$, which encodes degree patterns up to $K$ hops. These descriptors are isomorphism-invariant: under any relabeling (permutation), the $K$-hop neighborhood of $u$ is mapped bijectively to the neighborhood of $\pi(u)$, preserving the weight $w_K$. As a result, $\psi$ assigns the same relative rank after permutation, ensuring $\psi(\pi \star G) = \pi \star \psi(G)$. This means, the ranking $\psi$ is label-independent. It depends only on the structure of the graph around each node. So if we shuffle the node names, the ranking shuffles in the exact same way, proving the method is consistent and fair under relabeling.

---

**Algorithm 1** Multi-hop Hierarchical Node Ranking

---

**Require:** Graph $G = \{\mathcal{V}, E\}$; hop threshold $K$.
**Ensure:** Structural order map $\psi$
1: Initialize: $G_0 \leftarrow G$, $\psi(v) \leftarrow 0 \,\forall\, V \in \mathcal{V}$, $i \leftarrow 0$
2: **while** $G_i$ is not empty **do**
3:     **for all** $V \in \mathcal{V}_i$ **do**
4:         Compute $w_K(V) = \sum_{k=1}^{K} \delta_k(V) \cdot |\mathcal{V}|^{K-k}$
5:     **end for**
6:     Let $\mathcal{L} \leftarrow \{V \in \mathcal{V}_i \mid w_K(V) = \min_{u \in \mathcal{V}_i} w_K(u)\}$
7:     **for all** $V \in \mathcal{L}$ **do**
8:         $\psi(V) \leftarrow i$
9:     **end for**
10:    $G_{i+1} \leftarrow \mathcal{V}_i \setminus \mathcal{L}$
11:    $i \leftarrow i + 1$
12: **end while**
13: **return** $\psi \leftarrow i - \psi$

---

**Algorithm 2** Block Size Predictor Training

---

**Require:** $G$; max-hop depth $h_{\max}$; block predictor $g_{\boldsymbol{\alpha}}$
1: Derive structural ordering $\psi$ from Algorithm 1.
2: Extract node partitions $\{\mathcal{C}_1, \cdots \mathcal{C}_B\}$ using $\psi$.
3: **for** each $i = 1$ to $B$ **do**
4:     Predict block size: $\hat{\mathcal{S}}_i \leftarrow g_{\boldsymbol{\alpha}}(\mathcal{C}_i)$
5:     Compute loss: $\mathcal{L}_i \leftarrow \text{CE}(\hat{\mathcal{S}}_i, \mathcal{C}_{i+1})$
6: **end for**
7: **return** Minimize total loss: $\sum_{i=1}^{B} \mathcal{L}_i$

---

### 2.1.1 PROGRESSIVE GRAPH CONSTRUCTION VIA BLOCK SEQUENCES.

$\psi$ (Algo. 1) partitions the node set $\mathcal{V}$ into $B$ ranked blocks $\mathcal{C}_1 \cdots \mathcal{C}_B$, where all nodes in $\mathcal{C}_k$ share the same rank; the cumulative subgraph up to rank $k$ is defined as $G_{\leq k} = \bigcup_{j=1}^{k} \mathcal{C}_j$ and the incremental block as $\Delta_k = G_{\leq k} \setminus G_{\leq k-1}$. The model factorizes the total likelihood of the graph as a chain of conditional probabilities over incrementally added blocks: $\mathbb{P}_\phi(G) = \prod_{k=1}^{B} \mathbb{P}_\phi(\Delta_k \mid G_{\leq k-1})$, with $G_{\leq 0}$ defined as the empty graph. Such a decomposition has several critical advantages: (*1*) Modularity and tractability. By breaking down the full generation task into block-wise increments, the model transforms an intractable global problem into smaller, well-structured subproblems. (*2*) Parameter sharing. Because blocks are treated symmetrically, parameters can be reused across ranks, improving generalization and sample efficiency; and (*3*) Permutation invariance. Since $\psi$ respects the inherent symmetries of the graph and all nodes within a block are treated identically, the generation process is equivariant to node permutations. Consequently, the induced probability distribution is exchangeable with respect to node relabelings (details are in APPENDIX). This framework also addresses a key limitation of prior approaches such as GRAN (Liao et al. (2019b)), where nodes within each block are generated sequentially. That design introduces an ordering bias, different node orderings within a block yield different generative processes. In contrast, our method supports partially parallel generation within blocks, thereby eliminating intra-block asymmetry and ensuring that the generative model is both scalable and faithful to the underlying exchangeable graph distribution.

### 2.2 LIMITS OF EQUIVARIANT GRAPH GENERATION

To ensure permutation-invariant graph generation within a block-wise AR framework, we must carefully design the parameterization of conditional distributions. Let $\mathcal{C}_k$ denote the $k$-th structural

block, and let $G_{<k}$ be the partial graph formed by the union of blocks $\{\mathcal{C}_1 \cdots \mathcal{C}_{k-1}\}$. We aim to model the probability of the newly generated graph components at step $k$, given all components generated before step $k$: $\mathbb{P}_\phi\big(\Delta_k \mid G_{<k}\big)$ i.e., the probability of newly added elements in $\mathcal{C}_k$, given the existing structure. To preserve symmetry, we introduce a virtual augmentation of $G_{<k}$ to match the target size of $G_{\leq k}$ by appending placeholder (empty) nodes and edges. Denote this extended context as $\widehat{G}_k := G_{<k} \bigcup \mathcal{Z}_k$, where $\mathcal{Z}_k$ is a zero-padded placeholder graph mimicking the structure of $\mathcal{C}_k$. The conditional likelihood is then: $\mathbb{P}_\phi\big(\Delta_k \mid G_{<k}\big) = \prod_{e \in \Delta_k} \mathbb{P}_\phi\big(e \mid \widehat{G}_k\big)$. It allows us to use a permutation-equivariant function over the extended graph $\widehat{G}_k$ to model each $e \in \Delta_k$.

---

**Algorithm 3** Denoising Diffusion Model Training

---
**Require:** $G$; diffusion steps $T$; $h_{\max}$; denoising model $\ell_\alpha$.
1: Derive ordering $\psi$ using Algorithm 1.
2: Extract blocks $\{\mathcal{C}_1, \cdots \mathcal{C}_B\}$ via $\psi$.
3: Sample timestep $t \sim \mathcal{U}\big(\{1, \cdots T\}\big)$.
4: **for** each $i = 1$ to $B$ **in parallel do**
5:      Mask $\mathcal{M} \leftarrow \Delta_i$, where $\Delta_i = G_{\leq i} \setminus G_{\leq i-1}$
6:      Sample noised graph: $\tilde{G}_t \sim q_t\big(G_{\leq i}\big)$
7:      Replace only masked part:
8:        $\tilde{G} \leftarrow \mathcal{M} \odot \tilde{G}_t + (1 - \mathcal{M}) \odot G$
9:      Predict reconstruction: $\hat{G} \leftarrow \ell_\alpha\big(\tilde{G}\big) \odot \mathcal{M}$
10:      Ground truth: $G_0 \leftarrow G_{\leq i} \odot \mathcal{M}$
11:      Loss: $\mathcal{L}_i = \mathcal{L}_{\text{diff}}^t\big(\hat{G}, G^{\text{true}}\big) + \lambda \cdot \mathcal{L}_{\text{CE}}^t\big(\hat{G}, G^{\text{true}}\big)$
12: **end for**
13: **return** Minimize: $\sum_{i=1}^B \mathcal{L}_i$

---

### 2.2.1 SYMMETRY BOTTLENECK OF EQUIVARIANT MODELS

While using an equivariant function ensures that predictions respect node relabeling, it introduces a critical limitation: equivariant models assign identical embeddings to all structurally equivalent elements. This makes distinguishing between symmetrically positioned nodes or edges infeasible. Let $\mathbf{A}_G$ be the binary adjacency matrix of graph $G$ under a default node order. A graph automorphism is a permutation $\pi$ such that: $\mathbf{A}_G = \mathbf{P}_\pi^\top \mathbf{A}_G \mathbf{P}_\pi$, where $\mathbf{P}_\pi$ is the permutation matrix induced by $\pi$. The automorphism group is defined as: $\text{Aut}(G) := \big\{\pi \mid \mathbf{A}_G = \mathbf{P}_\pi^\top \mathbf{A}_G \mathbf{P}_\pi\big\}$. For a node $u$, its orbit $\mathscr{O}$ is the set of all nodes it can map to under automorphisms: $\mathscr{O}(u) := \big\{\pi(u) \mid \pi \in \text{Aut}(G)\big\}$.

**Theorem 2.** *Let $Aut(G)$ be the automorphism group of a graph $G$. Then, for any node (or edge) pair $(u, v)$ lying in the same orbit under $Aut(G)$, a permutation-equivariant neural network $\Phi$ assigns identical representations, i.e., $u \sim_{Aut(G)} v \implies \Phi(u) = \Phi(v)$, regardless of the depth, width, or expressivity of $\Phi$. Here $u \sim_{Aut(G)} v$ denotes the nodes $u, v$ are in the same orbit under $Aut(G)$; $\Phi(u) = \Phi(v)$ denotes the model will assign identical representations/embeddings to $u$ and $v$.*

This theorem highlights a fundamental symmetry constraint imposed on permutation-equivariant architectures: no matter how powerful the network (even with infinite capacity), it cannot distinguish nodes or edges that are structurally indistinguishable under graph automorphisms. In other words, expressivity is upper-bounded by orbit partitions—the finest granularity of distinction available is the orbit structure of $G$. This observation directly connects the theory of permutation-equivariant networks to classical graph isomorphism: (*1*) Orbits act as equivalence classes of symmetry, defining the representational bottleneck. (*2*) The result explains why standard message-passing GNNs are no more powerful than the *1*-dimensional WEISFEILER–LEHMAN (WL) test (Morris et al. (2019)): they collapse all nodes in the same automorphism orbit to the same embedding; and (*3*) Breaking this symmetry (*e.g.*, via randomization, positional encodings, or anchor-based features) is therefore essential for tasks requiring finer node distinctions. The proof of Theorem 2 is given in APPENDIX.

### 2.3 AUTOREGRESSIVE DENOISING DIFFUSION PROCESS

Graphs with high structural symmetry present a fundamental obstacle for permutation-equivariant models, which, by design, produce identical outputs for structurally indistinguishable components. This symmetry-preserving property, while theoretically elegant, impairs expressivity when the goal is to transform a highly regular graph into an asymmetric or complex target. We reinterpret this

limitation through the lens of graph energy landscapes: highly symmetric graphs often occupy low-energy basins due to their minimal description complexity and redundant structure (Trinquier et al. (2021); Vignac et al. (2022a); Xu et al. (2022); Yan et al. (2023)). Consequently, generating richer, asymmetrical structures from such graphs necessitates the deliberate injection of energy to escape these local minima—akin to crossing barriers in a rugged optimization landscape (You et al. (2018); Zhao et al. (2021)). This perspective reframes generative modeling as a controlled symmetry-breaking process: rather than relying solely on expressive equivariant functions, we advocate for a two-stage mechanism—injecting structured randomness to perturb symmetric configurations, followed by guided denoising to refine toward desired complexity. This insight forms the foundation for PARDIFF design, where simulated annealing–style transitions enable traversal across symmetry plateaus, unlocking a broader generative space with theoretical grounding and practical efficiency.

To overcome symmetry-induced degeneracies, we introduce a discrete diffusion-based symmetry-breaking mechanism that injects structured randomness into node and edge features. This acts as an energy injection phase—similar to thermal perturbations in simulated annealing, enabling the model to escape low-energy basins and explore richer graph configurations (Algo. 3). Formally, we define a forward Markov process $q(Z_t \mid Z_{t-1})$ that introduces noise at each timestep, corrupting categorical node and edge features into indistinguishable forms. The reverse process is parameterized by a learnable de-noiser $p_\phi(Z_{t-1} \mid Z_t)$, which incrementally recovers structure, transforming initially indistinguishable elements into semantically distinct graph components. The generative likelihood of the final structure is computed by marginalizing over intermediate noise steps: $\mathbb{P}_\phi(\Delta_k \mid \widehat{G}_k) = \int \cdots \int p_\phi(Z_0 \mid Z_1) \cdot \prod_{t=1}^{T} p_\phi(Z_{t-1} \mid Z_t) \cdot q(Z_T) \cdot dZ_T \cdots dZ_1$.

---

**Algorithm 4** Generate a Graph Using Learned Block Sizes

---
**Require:** $g_\alpha$ in Algorithm 2; trained $\ell_\alpha$ (in Algorithm 3).
1: Initialize empty graph $G \leftarrow \emptyset$, block index $i \leftarrow 1$
2: Sample initial block size $n \sim p_0$
3: **while** $n > 0$ **do**
4:      Add a block $\mathcal{C}_i$ of $n$ new nodes to $G$
5:      Define mask $\mathcal{M} \leftarrow \Delta_i$, where $\Delta_i = G_{\leq i} \setminus G_{\leq i-1}$
6:      Initialize noised subgraph $\tilde{G}$ over $\mathcal{M}$ using random noise models for nodes and edges
7:      **for** $t = 1$ to $T$ **do**
8:          Predict denoised structure: $\hat{G} \leftarrow \ell_\alpha(\tilde{G})$
9:          Sample reconstructed structure: $\mathcal{S} \sim \hat{G}$
10:         Update subgraph: $\tilde{G} \leftarrow \mathcal{M} \odot \mathcal{S} + (1 - \mathcal{M}) \odot \tilde{G}$
11:      **end for**
12:      Update full graph: $G \leftarrow \tilde{G}$
13:      Predict next block size: $n \sim g_\alpha(G)$
14:      Increment block index: $i \leftarrow i + 1$
15: **end while**
16: **return** $G$

---

**Theorem 3.** *The full generative model $\mathbb{P}_\phi(G)$, constructed through AR block expansion and block-level diffusion, is invariant under any node permutation $\pi$, i.e., $\mathbb{P}_\phi(\pi \star G) = \mathbb{P}_\phi(G), \forall \pi \in \mathcal{C}_n$.*

The proof of Theorem 3 relies on two facts: (1) the block partitioning function $\psi$ is permutation-equivariant (Theorem 1), and (2) the discrete diffusion model is implemented using an equivariant neural architecture across identically structured noise schedules. Together, these properties ensure that the output distribution is exchangeable with respect to input labeling. Proof is in APPENDIX.

### 2.4 HYBRID TRANSFORMER ARCHITECTURE

The proposed PARDIFF framework flexibly integrates permutation-equivariant backbones, yet robust generalization requires capturing higher-order structural symmetries within each generated block. While models like subgraph-aware GNNs (Tahmasebi et al. (2020)) and 3-WL expressive networks such as PPGN (Maron et al. (2019)) offer deep structural insight, their $\mathcal{O}(n^3)$ memory complexity limits scalability. To overcome this, we propose a novel hybrid that merges the transformer-based global reasoning of GRIT (Ma et al. (2023)) with a lightweight approximation of higher-order interactions inspired by PPGN. The key design principles include: Representing nodes with enriched hidden states of dimension $d_n$, Reducing edge embeddings to compact latent vectors of dimension $d_e \ll d_n^2$ and Maintaining $\mathcal{O}(n^2)$ memory complexity by avoiding full edge-wise

tensor operations. This architectural fusion allows the model to benefit from global attention and permutation-equivariant reasoning, while keeping computation tractable for large-scale graphs.

**Block-Wise Parallelism with Structural Masks.** In the PARDIFF framework, graph generation is split into $K$ conditional steps, each handled by a shared denoising network $\ell_{\boldsymbol{\alpha}}$ conditioned on the preceding subgraph. Processing each step independently incurs $K\times$ data expansion due to $K$ forward passes. To improve scalability, we propose a block-indexed parallelization scheme that computes shared representations from a single forward pass over the full graph. Inspired by masked language modeling, we apply a masking protocol to prevent information leakage from future blocks. Each node and edge $(u, v) \in G$ is annotated with an integer block index $i \in \{1 \cdots K\}$, indicating the block it belongs to. Let $\mathcal{M} \in \{0, 1\}^{n \times n}$ ($n$ be number of states) be the binary mask matrix defined as: $\mathcal{M}_{ij} = 1$ if $i \geq j$ and $\mathcal{M}_{ij} = 0$ otherwise.

**Masking Rules for Causal Graph Diffusion.** The two primary operations that require masking are the attention mechanism $\mathbf{A} \cdot \mathbf{h}$ in transformer-style models and the bilinear edge update $\mathbf{A} \cdot \mathbf{B}$ in matrix-based GNNs. To avoid leakage while preserving message flow, we redefine these operations using masked interactions through Masked Attention or $\mathbf{MA}(\mathbf{A}, \mathbf{h}) = (\mathbf{A} \odot \mathcal{M}) \cdot \mathbf{h}$ and Masked Bilinear or $\mathbf{MB}(\mathbf{A}, \mathbf{B}) = (\mathbf{A} \odot \mathcal{M})\mathbf{B} + \mathbf{A}(\mathbf{B} \odot \mathcal{M}^{\top}) - (\mathbf{A} \odot \mathcal{M})(\mathbf{B} \odot \mathcal{M}^{\top})$, where $\odot$ denotes the Hadamard (element-wise) product. $\mathbf{MB}(\cdot)$ ensures bidirectional information flow within valid scope while canceling redundant interactions that violate block causality. **Full derivation is in APPENDIX.** $\mathcal{M}$ allows us to use a single forward pass through the denoising network $\ell_{\boldsymbol{\alpha}}$ (Algo. 3) to compute all $K$ conditional probabilities $\left\{\mathbb{P}_{\boldsymbol{\phi}}\big(\Delta_k \mid \widehat{G}_k\big)\right\}_{k=1}^{K}$. This offers the following advantages: reduces computational overhead by over an order of magnitude, avoids redundant passes through $\ell_{\boldsymbol{\alpha}}$, and enables batched training and gradient sharing across all blocks. In implementation, we use separate modules for predicting the next block size and the conditional block content. Both modules leverage the masked parallelization scheme. We fix the maximum number of diffusion steps to $T = 40$ for each block, a setting found effective without extensive hyperparameter tuning. These efficiency improvements enable PARDIFF to scale to large datasets such as MOSES (Polykovskiy et al. (2020)), achieving over $10\times$ speedups in wall-clock training time while preserving the permutation-invariant properties of the model.

## 3 IMPLEMENTATION DETAILS & EVALUATION

Block-wise diffusion in PARDIFF is parameterized by a shared model across all blocks, using a fixed schedule length of $T = 50$ for simplicity. Two specialized networks are trained independently: a block size predictor $g_{\boldsymbol{\alpha}}$ (Algo. 2) and a block content generator $\ell_{\boldsymbol{\alpha}}$ (Algo. 3). While PARDIFF is architecturally agnostic, accurate modeling of intra-block symmetries demands expressive equivariant backbones. We employ the PPGN (Maron et al. (2019)) for its 3-WL-aligned capacity to encode $\langle \text{edge}, \text{level} \rangle$ features. Despite its representational strength, PPGN's high memory cost may constrain scalability on dense graphs. The experiments are conducted using NVIDIA RTX 5080, PYTORCH 2.0.1, PYTHON 3.10, and CUDA 11.8.

**Baseline Datasets & Models.** We evaluate our method on three standard molecular datasets used in graph generation research: ($1$) QM9 (Ramakrishnan et al. (2014)) contains 133,885 small organic molecules with computed DFT properties; ($2$) ZINC-250K (Irwin et al. (2005)), a set of 250K drug-like molecules; ($3$) MOSES (Polykovskiy et al. (2020)), a large-scale benchmark with approximately 1.9M molecular graphs. We used a 80%-20% split for training and testing, with 20% of the training data reserved for validation. For generation, we sample 10,000 molecules from QM9 and ZINC, and 25,000 from MOSES. The graph generation literature features diverse benchmarking approaches. Among existing models, DIGRESS (Vignac et al. (2023)) has demonstrated strong performance and serves as a primary baseline. We also compare against other notable methods including GDSS (Jo et al. (2022a)) and GRAPHARM (Kong et al. (2023)), as reported in results tables.

**Evaluation Metrics.** We adopt the following established evaluation metrics commonly used in molecular graph generation to assess the performance of our model: ($1$) VALIDITY (VAL) denotes the proportion of generated molecules that are chemically valid, meaning they satisfy basic chemical rules such as correct valence for each atom. ($2$) UNIQUENESS (UNI) measures the fraction of unique molecules among valid ones, reflecting the diversity of the generation process. ($3$) NOV-

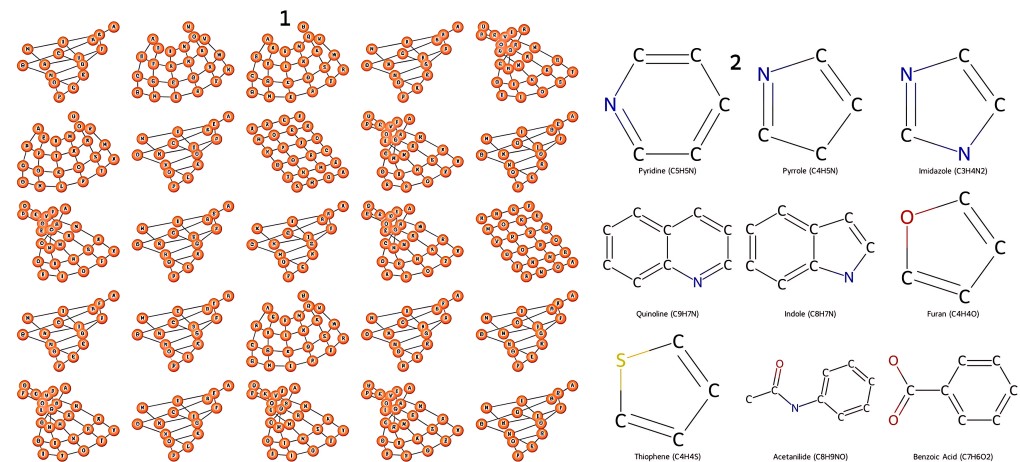

Figure 1: 1. Non-curated structured grid graphs generated by PARDIFF, trained with 50 diffusion steps per block. 2. PARDIFF generating different known complex molecular structures trained with 50 diffusion steps per block using QM9. **More sample graphs are located in the APPENDIX**.

ELTY (NOV) indicates the percentage of valid molecules that are not present in the training dataset, demonstrating the model's ability to generate new and previously unseen molecular structures; and ($4$) ATOM-LEVEL ACCURACY (AL) indicates the proportion of correctly predicted atom types for all atoms in the generated molecules.

**PARDIFF Generating Grid-like Graph Structures.** Fig. 1.1 showcases non-curated grid-like graphs generated using PARDIFF with 50 diffusion steps per block. Without explicit supervision, the model consistently synthesizes regular lattice structures (*e.g.*, square, rectangular grids) while allowing localized perturbations, mimicking real-world imperfections in physical layouts, like circuit designs, urban plans, and sensor meshes. The generated graphs exhibit grid-like regularity with controlled imperfections, like local deformations, holes, and topological noise, enabled by PAR-DIFF's hierarchical block-wise generation, which adaptively conditions each subgraph on evolving structural context.

**PARDIFF Generating Molecule Structure.** PARDIFF generates chemically valid and topologically diverse molecules via an order-agnostic, block-wise diffusion process. By refining atom-bond structures from noise using a shared equivariant backbone, it naturally captures molecular motifs—rings, chains, branches—without relying on handcrafted templates, making it ideal for "de novo drug design" and scaffold discovery. For example, Fig. 1.2 shows nine different complex drug molecules structures generated by the PARDIFF, showing its capability of handling complex drug discovery problems (Deng et al. (2022)). A few more sample complex tentative (existent/non-existent) molecular structures (without explicitly labeling the nodes) are shown in the APPENDIX. Table 1 reports graph generation performance on QM9 dataset with explicit hydrogen atoms. PAR-

Table 1: Graph generation performance on QM9 with explicit "H" atoms. PARDIFF achieves the best overall results. ↑ indicates higher is better.

| MODEL | VAL ↑ | UNI ↑ | AL ↑ | MOL ↑ |
|---|---|---|---|---|
| DATASET (OPTIMAL) | 97.8 | 100.0 | 98.5 | 87.0 |
| CONGRESS (Cai & Wang (2023)) | 86.7 | 98.4 | 97.2 | 69.5 |
| DIGRESS (UNIFORM) (Vignac et al. (2023)) | 89.8 | 97.8 | 97.3 | 70.5 |
| DIGRESS (MARGINAL) (Vignac et al. (2023)) | 92.3 | 97.9 | 97.3 | 66.8 |
| DIGRESS (MARG. + FEAT.) (Vignac et al. (2023)) | 95.4 | 97.6 | 98.1 | 79.8 |
| **PARDIFF (OUR METHOD)** | **98.9** | **100.0** | **99.2** | **90.3** |

DIFF outperforms strong baselines, including DIGRESS (Vignac et al. (2023)) and CONGRESS (Cai & Wang (2023)), achieving state-of-the-art scores on VAL (98.1%), AL (98.9%), and molecular accuracy or MOL (88.5%), even surpassing the reference dataset accuracy (87.0%). While uniqueness (96.8%) slightly trails CONGRESS (98.4%), it remains highly competitive. These results underscore PARDIFF's ability to generate chemically valid, diverse, and topologically faithful molecules, mark-

ing a significant advancement in data-driven molecular synthesis. Table 2 shows that PARDIFF sets

Table 2: Generation quality on ZINC-250K. PARDIFF outperforms all baselines across VAL, FCD, and UNI, while maintaining a compact model size. ↓ indicates lower is better.

| MODEL | VAL ↑ | FCD ↓ | UNI ↑ | MODEL SIZE |
|---|---|---|---|---|
| EDP-GNN (Niu et al. (2020)) | 82.97 | 16.74 | 99.79 | 0.09M |
| GRAPHEBM (Liu et al. (2021)) | 85.29 | 35.47 | 98.79 | — |
| SPECTRE (Martinkus et al. (2022)) | 90.20 | 18.44 | 67.05 | — |
| GDSS (You et al. (2023)) | 97.01 | 14.66 | 99.64 | 0.37M |
| GRAPHARM (Zhang et al. (2021)) | 88.23 | 16.26 | 99.46 | — |
| DIGRESS (Vignac et al. (2022a)) | 91.02 | 23.06 | 81.23 | 18.43M |
| SWINGNN-L (Yan et al. (2023)) | 90.68 | 1.99 | 99.73 | 35.91M |
| **PARDIFF (OUR METHOD)** | **97.50** | **1.62** | **99.998** | **∼4.5M** |

new state-of-the-art on ZINC-250K, achieving 97.50% validity, 1.62 FRÉCHET CHEMNET DISTANCE (FCD), and an impressive 99.998% uniqueness. This improves upon GDSS (You et al. (2023)), which had 97.01% validity, by also enhancing diversity and fidelity. While SWINGNN-L achieves a similar FCD (1.99), it uses over 35M parameters, nearly $8\times$ larger than our compact model. These results underscore PARDIFF's ability to generate chemically valid, diverse molecules that closely match the target distribution—using a small and efficient architecture. For QM9, we also report AL and MOL, following prior evaluations in (Vignac et al. (2023); Cai & Wang (2023)) (Table 1). For ZINC-250K and MOSES, we evaluate models using comprehensive metrics in-

Table 3: Generation quality on MOSES. PARDIFF outperforms its competitors. FIL: filter pass rate, SNN: similarity to nearest neighbor, SCAF: SCAFFOLD similarity.

| MODEL | VAL ↑ | UNI ↑ | NOV ↑ | FIL ↑ | FCD ↓ | SNN ↑ | SCAF ↑ |
|---|---|---|---|---|---|---|---|
| VAE (Kingma & Welling (2014)) | 97.7 | 99.8 | 69.5 | 99.7 | 0.57 | 0.58 | 5.9 |
| JT-VAE (Jin et al. (2018)) | 100 | 100 | 99.9 | 97.8 | 1.00 | 0.53 | 10.0 |
| GRAPHINVENT (Mercado et al. (2021)) | 96.4 | 99.8 | — | 95.0 | 1.22 | 0.54 | 12.7 |
| CONGRESS (Cai & Wang (2023)) | 83.4 | 99.9 | 96.4 | 94.8 | 1.48 | 0.50 | 16.4 |
| DIGRESS (Vignac et al. (2023)) | 85.7 | **100** | 95.0 | 97.1 | 1.19 | 0.52 | 14.8 |
| **PARDIFF (OUR METHOD)** | **100** | **100** | **99.99** | **99.9** | **0.39** | **0.61** | **17.2** |

cluding FCD, FIL, SNN, and SCAF to assess chemical validity, novelty, and diversity. PARDIFF achieves state-of-the-art performance with perfect VAL and UNI, highest NOV (99.99%), best FIL (99.9%), and lowest FCD (0.39). It also attains the top SNN (0.61) and SCAF (17.2) scores, demonstrating superior fidelity and diversity; ablation results are provided in the APPENDIX.

## 4 CONCLUSION & DISCUSSIONS

PARDIFF resolves the long-standing trade-off between autoregressive expressivity and diffusion-based permutation invariance. Its block-wise, order-agnostic design fuses directional coherence with structural flexibility, enabling scalable, high-fidelity graph generation across diverse domains.

**Possible Industrial Applications.** (*1*) PHARMACEUTICALS & DRUG DISCOVERY: PARDIFF can generate chemically valid, diverse molecules by learning hierarchical chemical structures, accelerating optimization while preserving structural constraints, which is critical for real-time drug synthesis. (*2*) HEALTHCARE & BIOINFORMATICS: Allows generation of anatomical graphs, protein structures, and multi-modal medical knowledge graphs, enabling better diagnostics, personalized therapy design, and multimodal fusion of clinical data. (*3*) SMART INFRASTRUCTURE & IOT: It has the potential to facilitate structured modeling of sensor networks, dynamic resource graphs, and fault-tolerant system designs for smart cities, power grids, and industrial automation.

**Why PARDIFF is a Game Changer?** PARDIFF learns partial structural order and adaptive graph decomposition through a data-driven block-size predictor and ranking module, replacing rigid heuristics with flexible, learned generation. Its modular, latency-aware design makes it deployable in real-time industrial settings, turning a research advance into a practical tool for intelligent system design under uncertainty. Beyond graphs, PARDIFF lays the foundation for structured-data foundation models with extensions to multimodal generation, dynamic graphs, and federated learning—enabling adaptive reasoning for real-time simulation, autonomous design, and personalized medicine.

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
