# A APPENDIX/SUPPLEMENTAL MATERIAL

## A.1 USED SYMBOLS

Table 4: Explanation of the symbols used in the paper.

| Used Symbols | Descriptions |
|---|---|
| $G_0 = \{\mathcal{V}_0, E_0\}$ | Clean graph |
| $\mathcal{V}_t : t = 1 \cdots T$ | One-hot encoded node-$i$ features |
| $E_t : t = 1 \cdots T$ | One-hot encoded edge features |
| $q(G_t \mid G_{t-1})$ | Noise-driven forward diffusion process |
| $p_\phi(G_{t-1} \mid G_t)$ | Reverse diffusion or denoising |
| $n$ | Number of states |
| $\mathcal{C}_i : i = 1 \cdots B$ | $i$-th block containing nodes |
| $G_{\leq k}$ | $\mathcal{C}_1 \bigcup \cdots \bigcup \mathcal{C}_k.$ |

## A.2 VARIATIONAL OBJECTIVE FOR STRUCTURED GRAPH DIFFUSION

To train the reverse denoising model $p_\theta(G_{t-1} \mid G_t)$ to approximate the true posterior of the forward process $q_\phi(G_{t-1} \mid G_t, G_0)$, we derive a variational lower bound (VLB) on the marginal likelihood $\log p_\theta(G_0)$. Starting from the evidence lower bound:

$$\log p_\theta(G_0) = \log \int q_\phi(G_{1:T} \mid G_0) \cdot \frac{p_\theta(G_0, G_{1:T})}{q_\phi(G_{1:T} \mid G_0)} \, dG_{1:T} \geq \mathbb{E}_{q_\phi(G_{1:T}|G_0)} \left[ \log \frac{p_\theta(G_0, G_{1:T})}{q_\phi(G_{1:T} \mid G_0)} \right] \tag{3}$$

We decompose the joint distributions as:

$$p_\theta(G_0, G_{1:T}) = p_\theta(G_T) \prod_{t=2}^{T} p_\theta(G_{t-1} \mid G_t),$$

$$q_\phi(G_{1:T} \mid G_0) = \prod_{t=1}^{T} q_\phi(G_t \mid G_{t-1}, G_0) \tag{4}$$

Substituting into the ELBO:

$$\log p_\theta(G_0) \geq \mathbb{E}_{q_\phi} \left[ \log p_\theta(G_T) + \sum_{t=2}^{T} \log p_\theta(G_{t-1} \mid G_t) - \sum_{t=1}^{T} \log q_\phi(G_t \mid G_{t-1}, G_0) \right] \tag{5}$$

This can be rearranged as:

$$\log p_\theta(G_0) \geq \mathbb{E}_{q_\phi} \left[ \log p_\theta(G_T) - \log q_\phi(G_1 \mid G_0) + \sum_{t=2}^{T} \log \frac{p_\theta(G_{t-1}|G_t)}{q_\phi(G_t|G_{t-1}, G_0)} \right] \tag{6}$$

We reorganize the objective into reconstruction and KL terms:

$$\log p_\theta(G_0) \geq \mathbb{E}_{q_\phi} \left[ \log p_\theta(G_0 \mid G_1) \right] - \sum_{t=2}^{T} \mathbb{E}_{q_\phi} \left[ D_{\mathrm{KL}} \left( q_\phi(G_{t-1} \mid G_t, G_0) \, \| \, p_\theta(G_{t-1} \mid G_t) \right) \right]$$
$$- \, \mathrm{const.} \tag{7}$$

We define the total training objective as:

$$\mathcal{L}(\theta) = \mathcal{L}_{\mathrm{rec}}(\theta) + \sum_{t=2}^{T} \mathcal{L}_t(\theta) \tag{8}$$

where:

$$\mathcal{L}_{\text{rec}}(\theta) = -\mathbb{E}_{q_\phi} \left[ \log p_\theta(G_0 \mid G_1) \right] \tag{9}$$

$$\mathcal{L}_t(\theta) = \mathbb{E}_{q_\phi} \left[ D_{\text{KL}} \left( q_\phi(G_{t-1} \mid G_t, G_0) \,\|\, p_\theta(G_{t-1} \mid G_t) \right) \right] \tag{10}$$

This variational bound enables efficient training via a hybrid loss that balances data likelihood with forward–reverse consistency across diffusion steps.

### A.3 PARAMETERIZING FORWARD AND REVERSE TRANSITIONS IN DISCRETE GRAPH DIFFUSION

We define the forward and reverse diffusion processes over graphs using a simplified discrete-time formulation, following Zhao et al. Zhao et al. (2024). Our framework focuses on three key distributions: (i) the forward marginal $q(G_t \mid G_0)$, (ii) the backward posterior $q(G_{t-1} \mid G_t, G_0)$, and (iii) the learned reverse process $p_\phi(G_{t-1} \mid G_t)$. This design prioritizes memory efficiency and avoids the complexity introduced by approximations such as those in D3PM Austin et al. (2021).

Since the forward process applies noise independently to all nodes and edges (as shown in Eq. (1)), we can model these three distributions by factorizing over individual elements. Let $x \in \mathcal{V}_t \cup \mathcal{E}_t$ be a discrete random variable with one-hot encoding and categorical distribution: $x \sim \text{Cat}(x; \mathbf{p})$, where $\mathbf{p} \in [0,1]^n$ and $\mathbf{1}^\top \mathbf{p} = 1$. Then the probability of observing a one-hot state $x$ under distribution $\mathbf{p}$ is $x^\top \mathbf{p}$. The corruption step $q(x_t \mid x_{t-1})$ can be expressed using a transition matrix $Q_t \in [0,1]^{n \times n}$ as:

$$q(x_t \mid x_{t-1}) = \text{Cat}(x_t; Q_t^\top x_{t-1}) \tag{11}$$

Let the composed transition matrix be $\overline{Q}_t = Q_1 Q_2 \cdots Q_t$. Then, the forward marginal becomes:

$$q(x_t \mid x_0) = \text{Cat}(x_t; \overline{Q}_t^\top x_0) \tag{12}$$

The backward posterior is:

$$q(x_{t-1} \mid x_t, x_0) = \text{Cat} \left( x_{t-1}; \frac{Q_t x_t \odot \overline{Q}_{t-1}^\top x_0}{x_t^\top \overline{Q}_t^\top x_0} \right) \tag{13}$$

This applies to both node states $x = v_i^t \in \mathcal{V}_t$ and edge types $x = e_{i,j}^t \in \mathcal{E}_t$, with the same formulation. We optionally use shared transition matrices $Q_t^{\mathcal{V}}$ and $Q_t^{\mathcal{E}}$ for all nodes and edges, respectively. See the next section for derivation. To define a uniform and information-less terminal distribution $q(G_T \mid G_0)$, we choose:

$$Q_t = \alpha_t I + (1 - \alpha_t) \mathbf{1} \mathbf{m}^\top \tag{14}$$

where $\alpha_t \in [0,1]$ is a time-dependent noise schedule, and $\mathbf{m} \in [0,1]^n$ is the uniform categorical distribution over $n$ states, such that $m_i = \frac{1}{n}$. For reverse modeling, we use:

$$p_\phi(x_{t-1} \mid G_t) = \sum_{x_0} q(x_{t-1} \mid x_t, x_0) \cdot p_\phi(x_0 \mid G_t) \tag{15}$$

This formulation enables us to parameterize $p_\phi(x_0 \mid G_t)$ with a neural network and compute $p_\phi(x_{t-1} \mid G_t)$ via marginalization over the clean state space using Eq. (3).

### A.4 DERIVATION OF $q(x_{t-1} \mid x_t, x_0)$

We begin by defining the composite transition matrix over steps $s$ through $t$ as $\overline{Q}_{t|s} = Q_s Q_{s+1} \cdots Q_t$. For brevity, we denote $\overline{Q}_t = \overline{Q}_{t|0}$ and $\overline{Q}_{t-1} = \overline{Q}_{t-1|0}$. Our goal is to compute the posterior distribution $q(x_{t-1} \mid x_t, x_0)$, assuming the forward process has the Markov structure given by:

$$q(x_t \mid x_{t-1}) = \text{Cat}(x_t; Q_t^\top x_{t-1}) \tag{16}$$

From the chain rule of probability:

$$q(x_{t-1} \mid x_t, x_0) = \frac{q(x_t \mid x_{t-1})q(x_{t-1} \mid x_0)}{q(x_t \mid x_0)} \tag{17}$$

We expand each term using the forward marginals:

$$q(x_{t-1} \mid x_0) = \text{Cat}(x_{t-1}; \overline{Q}_{t-1}^{\top} x_0) \tag{18}$$

$$q(x_t \mid x_0) = \text{Cat}(x_t; \overline{Q}_t^{\top} x_0) \tag{19}$$

Thus, the numerator becomes:

$$q(x_t \mid x_{t-1})q(x_{t-1} \mid x_0) = (x_t^{\top} Q_t^{\top} x_{t-1}) \cdot (x_{t-1}^{\top} \overline{Q}_{t-1}^{\top} x_0)$$

We now marginalize over all possible $x_{t-1}$ to normalize:

$$q(x_{t-1} \mid x_t, x_0) = \frac{Q_t x_t \odot \overline{Q}_{t-1}^{\top} x_0}{x_t^{\top} \overline{Q}_t^{\top} x_0}$$

Hence, the posterior is a categorical distribution over $x_{t-1}$:

$$q(x_{t-1} \mid x_t, x_0) = \text{Cat}\left(x_{t-1}; \frac{Q_t x_t \odot \overline{Q}_{t-1}^{\top} x_0}{x_t^{\top} \overline{Q}_t^{\top} x_0}\right) \tag{20}$$

### A.5 Proof of Permutation Invariance in Blockwise Graph Generation

To establish that $p_\theta(G)$ is an exchangeable probability distribution over graphs, we aim to prove that for any permutation matrix $P$, the model satisfies

$$p_\theta(P \star G) = p_\theta(G), \tag{21}$$

where $P \star G$ denotes the graph obtained by permuting both node indices and corresponding edge entries in $G$.

Our generative model factorizes the likelihood of a graph $G$ based on block-wise decomposition induced by a structural ranking function $\psi$. Let $\mathcal{B}_1, \ldots, \mathcal{B}_{K_B}$ be the node subsets (blocks) ranked by $\psi$. The generation is performed sequentially over these blocks:

$$p_\theta(G) = \prod_{i=1}^{K_B} p_\theta\left(G[\mathcal{B}_i] \mid G[\mathcal{B}_{1:i-1}], G[\mathcal{B}_{1:i-1}] \setminus G[\mathcal{B}_i]\right). \tag{22}$$

**Permutation Equivariance of the Indexing.**   Each block $\mathcal{B}_i$ is determined from $G$ using $\psi(G)$, which is permutation-consistent (Theorem 1). Thus, for any permutation matrix $P$, we have

$$\mathcal{B}_i(P \star G) = P \star \mathcal{B}_i(G). \tag{23}$$

Furthermore, indexing operations on graphs are equivariant:

$$P \star G[\mathcal{B}_i] = G[P \star \mathcal{B}_i]. \tag{24}$$

**Exchangeability of Block Generation.**   Consider:

$$p_\theta(P \star G) = \prod_{t=1}^{K_B} p_\theta\left(P \star G[\mathcal{B}_t] \mid P \star G[\mathcal{B}_{1:t-1}],\right.$$

$$\left. P \star (G[\mathcal{B}_{1:t-1}] \setminus G[\mathcal{B}_t])\right) \tag{25}$$

Since our model is constructed to be equivariant with respect to permutations, each conditional satisfies:

$$p_\theta(P \star X \mid P \star Y) = p_\theta(X \mid Y), \tag{26}$$

for arbitrary subgraphs $X, Y$. Applying this recursively yields:

$$p_\theta(P \star G) = \prod_{i=1}^{K_B} p_\theta\left(G[\mathcal{B}_i] \mid G[\mathcal{B}_{1:i-1}], G[\mathcal{B}_{1:i-1}] \setminus G[\mathcal{B}_i]\right) = p_\theta(G). \tag{27}$$

**Marginalization of Conditioning Sets.** For further rigor, define the conditional term

$$p_\theta \left( G[\mathcal{B}_i] \mid G[\mathcal{B}_{1:i-1}] \setminus G[\mathcal{B}_i], G[\mathcal{B}_{1:i-1}] \right). \tag{28}$$

Let $\mathcal{H}_{\mathcal{B}_{1:i-1}}$ denote all other nodes outside $\mathcal{B}_{1:i}$. Then,

$$p_\theta \left( G[\mathcal{B}_i] \mid G[\mathcal{B}_{1:i-1}], G[\mathcal{B}_{1:i-1}] \setminus G[\mathcal{B}_i] \right)$$

$$= \int p_\theta \left( G[\mathcal{B}_i] \mid G[\mathcal{B}_{1:i-1}], \mathcal{H}_{\mathcal{B}_{1:i-1}} \right) \cdot p(\mathcal{H}_{\mathcal{B}_{1:i-1}}) \, d\mathcal{H}_{\mathcal{B}_{1:i-1}}. \tag{29}$$

Using the fact that the generative model's forward noise and reverse denoising chains are designed to be permutation equivariant, we have:

$$p_\theta(G) = \int p(H_T|G) \prod_{t=1}^{T} p(H_{t-1}|H_t) \, dH_{1:T}. \tag{30}$$

Then for any $P$:

$$p_\theta(P \star G) = \int p(H_T|P \star G) \prod_{t=1}^{T} p(H_{t-1}|H_t) \, dH_{1:T}, \tag{31}$$

$$= \int p(P \star H_T|G) \prod_{t=1}^{T} p(P \star H_{t-1}|P \star H_t) \, dH_{1:T}, \tag{32}$$

$$= \int p(H_T|G) \prod_{t=1}^{T} p(H_{t-1}|H_t) \, dH_{1:T} = p_\theta(G). \tag{33}$$

This confirms that $p_\theta(G)$ is invariant under any node permutation $P$, establishing exchangeability.

### A.6 UNIFIED TRAINING AND GENERATION FOR BLOCK-WISE STRUCTURED GRAPH DIFFUSION

---

**Algorithm 5** Unified Training and Generation Procedure for Block-wise Structured Graph Diffusion

---

**Require:** Graph $G$, max diffusion steps $T$, max hop $K_h$, block size predictor $g_\theta$, denoising model $\ell_\alpha$
1: Obtain node ordering $\psi$ from ordering network $\phi$ (Algorithm 1)
2: Partition $G$ into ranked blocks $[\mathcal{C}_1, \ldots, \mathcal{C}_{K_B}]$ using $\psi$
3: **for** $i = 1$ to $K_B$ **do**
4:     $\widehat{\mathcal{C}}_i \leftarrow g_\theta(G_{\leq i-1})$
5:     $\mathcal{M} \leftarrow \text{mask}(G[\mathcal{C}_{1:i}] \setminus G[\mathcal{C}_{1:i-1}])$
6:     Sample $t \sim \mathcal{U}(1, T)$
7:     $\widetilde{G}[\mathcal{C}_i] \leftarrow \mathcal{M} \odot q_t(G[\mathcal{C}_i]) + (1 - \mathcal{M}) \odot G[\mathcal{C}_i]$
8:     $X \leftarrow f_\theta(\widetilde{G}[\mathcal{C}_i])$
9:     Compute $\ell_i^{\text{KL}}$ and $\ell_i^{\text{CE}}$ using Eq. (2), Eq. (3)
10: **end for**
11: Minimize total loss: $\sum_{i=1}^{K_B} \ell_i$
12: $G \leftarrow \emptyset, i \leftarrow 1$
13: Sample $n \sim g_\theta(G)$
14: **while** $n > 0$ **do**
15:     Add block $\mathcal{C}_i$ with $n$ nodes to $G$
16:     $\mathcal{M} \leftarrow \text{mask}(G[\mathcal{C}_i] \setminus G[\mathcal{C}_{1:i-1}])$
17:     $\widetilde{G} \leftarrow \text{Noise}(\mathcal{M})$
18:     **for** $j = 1$ to $T$ **do**
19:         $\mathbf{p} \leftarrow f_\theta(\widetilde{G})$
20:         Sample $S$ from $\mathbf{p}$
21:         $\widetilde{G} \leftarrow \mathcal{M} \odot S + (1 - \mathcal{M}) \odot \widetilde{G}$
22:     **end for**
23:     $G \leftarrow \widetilde{G}$
24:     Sample $n \sim g_\theta(G)$
25:     $i \leftarrow i + 1$
26: **end while**
27: **return** $G$

---

### A.7 Proof of Theorem 1

Consider a sequence of transition matrices $\left\{\mathscr{T}_1, \cdots \mathscr{T}_T\right\}$, each representing a categorical diffusion step. The matrices should be constructed such that, at long time horizons $(t \to T)$, the resulting distribution converges to a known steady-state distribution $\boldsymbol{\mu} \in D^K$, where $D^K$ is the $K$-dimensional probability simplex. We define this limiting behavior as:

$$\lim_{t \to T} \mathscr{T}_t = \mathbf{1}\boldsymbol{\mu}^\top \tag{34}$$

This ensures that every row of the composed matrix approaches $\boldsymbol{\mu}$, making the distribution stationary. To enforce this convergence in a controllable way, we propose defining each transition matrix $\mathscr{T}_t$ as a convex blend between the identity matrix and the rank-1 matrix $\mathbf{1}\boldsymbol{\mu}^\top$:

$$\mathscr{T}_t = \gamma_t \cdot \mathbf{I} + \left(1 - \gamma_t\right) \cdot \mathbf{1}\boldsymbol{\mu}^\top, \quad \gamma_t \in [0, 1] \tag{35}$$

The accumulated transition from time step $s$ to $t$, denoted as $\mathscr{T}_{t|s}$, can be recursively written as follows:

$$\mathscr{T}_{t|s} = \gamma_{t|s} \cdot \mathbf{I} + \left(1 - \gamma_{t|s}\right) \cdot \mathbf{1}\boldsymbol{\mu}^\top, \tag{36}$$

where the effective decay factor $\gamma_{t|s}$ is the product of all decay terms from step $s + 1$ to $t$:

$$\gamma_{t|s} = \prod_{r=s+1}^{t} \gamma_r \tag{37}$$

This implies:

$$\gamma_t = \gamma_{t|0} = \gamma_{t|s} \cdot \gamma_s \tag{38}$$

With this formulation, we ensure that as $t \to T$, the accumulated matrix $\mathscr{T}_{t|0}$ becomes fully rank-1, and the variable distribution becomes indistinguishable from the stationary prior $\boldsymbol{\mu}$. This gives the reparameterized posterior for timestep $t - 1$, used in computing the variational loss. We present the argument for node representations; the same reasoning holds for structurally symmetric edges. Suppose that *two* nodes $u$ and $v$ in a graph $G$ are structurally indistinguishable. Then, there exists a graph automorphism $\pi \in \text{Aut}(G)$ such that:

$$\pi(u) = v \tag{39}$$

Let $\mathcal{P}_n$ denote the set of all node permutation matrices of size $n \times n$. Assume we have a neural function $\psi : G \mapsto \mathbb{R}^{n \times d}$ that is permutation-equivariant, i.e., for any permutation matrix $\pi \in \mathcal{C}_n$, we have:

$$\psi\left(\pi \star G\right) = \pi \star \psi\left(G\right) \tag{40}$$

Now, apply $\pi$ as the permutation on nodes. Because $\pi$ is an automorphism of $G$, it preserves the graph structure, so $\pi \star G = G$. Thus:

$$\psi\left(G\right) = \psi\left(\pi \star G\right) = \pi \star \psi\left(G\right) \tag{41}$$

This implies:

$$\psi\left(G\right)_u = \psi\left(G\right)_v \tag{42}$$

In other words, nodes $u$ and $v$, being symmetric under graph automorphism $\pi$, are mapped to identical representations by the function $\psi$.

### A.8 Proof of Theorem 2

Let $\sigma \in \text{Auto}(G)$. By definition, $\sigma \star G = G$(the attributed graph is unchanged by $\sigma$). By permutation equivariance of $\Phi$,

$$\Phi(G) = \Phi(\sigma \star G) = \sigma \star \Phi(G). \tag{43}$$

Unpacking the rightmost equality component-wise over nodes gives, for every $w \in \mathcal{V}$,

$$\Phi(G)_w = \big(\sigma \star \phi(G)\big)_w = \phi(G)_{\sigma^{-1}(w)}. \tag{44}$$

Equivalently, for every $w$, $\Phi(G)_{\sigma(w)} = \Phi(G)_w$. Now fix any two nodes $u, v \in \mathcal{V}$ in the same $\mathrm{Aut}(G)$-orbit. By definition, there exists $\sigma \in \mathrm{Aut}(G)$ with $\sigma(u) = v$. Applying the relation above with $w = u$ yields

$$\Phi(G)_v = \phi(G)_{\sigma(u)} = \Phi(G)_u, \tag{45}$$

establishing $\Phi(G)_u = \Phi(G)_v$. The argument for edge embeddings is identical: let $\Phi^{(e)}$ map $G$ to edge-wise outputs indexed by ordered (or unordered) pairs. Equivariance acts on pairs via $\pi \star (i, j) = \big(\pi(i), \pi(j)\big)$. For any automorphism $\sigma$, $\Phi^{(e)}(G) = \sigma \star \Phi^{(e)}(G)$, hence $\Phi^{(e)}(G)_{(i,j)} = \Phi^{(e)}(G)_{(\sigma(i), \sigma(j))}$. If $(u, v)$ and $(u', v')$ are in the same orbit, choose $\sigma$ with $\sigma(u) = u'$, $\sigma(v) = v'$ to conclude equality of their edge embeddings. Since the derivation uses only (1) $\sigma \star G = G$ for $\sigma \in \mathrm{Aut}(G)$ and (2) equivariance of $\Phi$, the result is independent of depth/width/expressivity.

**Implication.** (1) On features. The statement assumes automorphisms preserve all attributes used by $\Phi$. If node/edge features break symmetry (e.g., unique IDs), then $\mathrm{Aut}(G)$ shrinks accordingly; the conclusion applies with respect to that reduced group. (2) Symmetry cannot be broken internally. The proof formalizes the impossibility of distinguishing nodes within an automorphism orbit by any permutation-equivariant architecture alone. To separate orbit-mates, one must inject symmetry-breaking signals (positional encodings, random IDs, anchors, or global tie-breakers); and (3) Group-theoretic view. The equality $\Phi(G) = \sigma \star \phi(G) \ \forall \ \sigma \in \mathrm{Aut}(G)$ means $\Phi(G)$ lies in the fixed-point subspace of the representation of $\mathrm{Aut}(G)$. Constancy on orbits is exactly the characterization of such fixed points by Burnside's lemma/orbit–stabilizer intuition.

## A.9    PROOF OF THEOREM 3

To demonstrate that the learned probability distribution $\mathbb{P}_\phi(G)$ over graphs is exchangeable, we must verify that for any node permutation matrix $\pi \in \mathcal{C}_n$, the group of node permutations, it holds that:

$$\mathbb{P}_\phi\big(\pi \star G\big) = \mathbb{P}_\phi\big(G\big) \tag{46}$$

Here, $\pi \star G$ denotes the permuted graph, where nodes and their relations (or, edges) are permuted accordingly: $\pi \star G = \big(\pi \cdot V, \pi \cdot E \cdot \pi^\top\big)$. Assume that the generation model produces a graph via a sequential composition of subgraphs defined by structural neighborhoods or partitions, such that:

$$\mathbb{P}_\phi\big(G\big) = \prod_{i=1}^{K} \mathbb{P}_\phi\big(G_{\leq i} \setminus G_{\leq i-1} \mid G_{\leq i-1}\big) \tag{47}$$

Here, $G_{\leq i}$ denotes the union of the first $i$ block $\mathcal{C}_1 \cdots \mathcal{C}_i$ (e.g., neighborhoods) induced by a binary mask over nodes. This indexing operation is permutation-equivariant:

$$\pi \star \big(G_{\leq i}\big) = \big(\pi \star G\big)_{\leq i} \tag{48}$$

Additionally, suppose that each subset $\big(\pi \star G\big)_{\leq i}$ is selected via a deterministic function of the graph structure (e.g., via neighborhood expansion or hop-based grouping), which is also equivariant under permutation. Then:

$$\big(\pi \star G\big)_{\leq i} = \pi \star G_{\leq i} \tag{49}$$

Now evaluate the generative model on the permuted graph:

$$\mathbb{P}_\phi\big(\pi \star G\big) = \prod_{i=1}^K \mathbb{P}_\phi\big((\pi \star G)_{\leq i} \setminus (\pi \star G)_{\leq i-1} \mid (\pi \star G)_{\leq i-1}\big)$$

$$= \prod_{i=1}^K \mathbb{P}_\phi\big(\pi \star \big(G_{\leq i} \setminus G_{\leq i-1}\big) \mid \pi \star G_{\leq i-1}\big) \tag{50}$$

$$= \prod_{i=1}^K \mathbb{P}_\phi\big(\pi \star \Delta_i \mid \pi \cdot \mathcal{G}_{<i}\big),$$

where $\Delta_i = G_{\leq k} \setminus G_{\leq i-1}$, and $G_{<i} = \mathcal{C}_1 \bigcup \cdots \bigcup \mathcal{C}_{i-1}$. If the conditional probabilities $\mathbb{P}_\phi$ are defined through permutation-invariant functions (e.g., based on multi-set or degree statistics), then we have:

$$\mathbb{P}_\phi\big(\pi \star \Delta_i \mid \pi \star G_{\leq i}\big) = \mathbb{P}_\phi\big(\Delta_i \mid G_{\leq i}\big) \tag{51}$$

Thus,

$$\mathbb{P}_\phi\big(\pi \star G\big) = \mathbb{P}_\phi\big(G\big), \tag{52}$$

which confirms the probability distribution modeled by $\mathbb{P}_\phi$ is invariant under node permutations, i.e., it is exchangeable.

## A.10 Deriving a Block-Causal Matrix Product

Let $X \in \mathbb{R}^{n \times d}$ and $Y \in \mathbb{R}^{d \times m}$ be two matrices. Define the standard matrix multiplication entry as:

$$\big[XY\big]_{ij} = \langle \mathbf{x}_i, \mathbf{y}_j \rangle, \tag{53}$$

where $\mathbf{x}_i$ denotes the $i$-th row of $X$, and $\mathbf{y}_j$ is the $j$-th column of $Y$. Now, in a block-wise AR setting, we introduce a function $b : \{1 \cdots n\} \mapsto \mathbb{N}$ assigning a block index to each row/column. The matrix entry $(i, j)$ should depend only on features from block indices $\leq \max\big(b(i), b(j)\big)$. To ensure this, define a binary mask matrix $\mathcal{M} \in \{0, 1\}^{n \times d}$, where:

$$\mathcal{M}_{ik} = \begin{cases} 1 & \text{if } b(i) \geq b(k) \\ 0 & \text{otherwise} \end{cases} \tag{54}$$

For a safe computation of the entry $\mathcal{Z}_{ij}$ under this constraint, we define:

$$\mathcal{Z}_{ij} = \langle \mathbf{x}_i \odot \big(\boldsymbol{\mu}_i \vee \boldsymbol{\mu}_j\big), \mathbf{y}_j \rangle \tag{55}$$

Here, $\boldsymbol{\mu}_i$ and $\boldsymbol{\mu}_j$ are binary indicator vectors selecting valid components, and $\odot$ denotes the Hadamard (element-wise) product, while $\vee$ is the element-wise logical OR. We can expand this expression as:

$$\mathcal{Z}_{ij} = \langle \mathbf{x}_i \odot \boldsymbol{\mu}_i, \mathbf{y}_j \rangle + \langle \mathbf{x}_i, \mathbf{y}_j \odot \boldsymbol{\mu}_j \rangle - \langle \mathbf{x}_i \odot \boldsymbol{\mu}_i, \mathbf{y}_j \odot \boldsymbol{\mu}_j \rangle \tag{56}$$

In matrix form, letting $\mathbf{Z}$ be the final output:

$$\mathbf{Z} = \big(\mathbf{X} \odot \mathcal{M}\big)\mathbf{Y} + \mathbf{X}\big(\mathbf{Y} \odot \mathcal{M}^\top\big) - \big(\mathbf{X} \odot \mathcal{M}\big)\big(\mathbf{Y} \odot \mathcal{M}^\top\big) \tag{57}$$

This formulation ensures that information flows only within valid block boundaries, enabling parallelizable yet causally consistent matrix computation.

## A.11 Structured Grid Graphs

This section presents a few more structured artificial grid generated using the proposed PARDIFF algorithm:

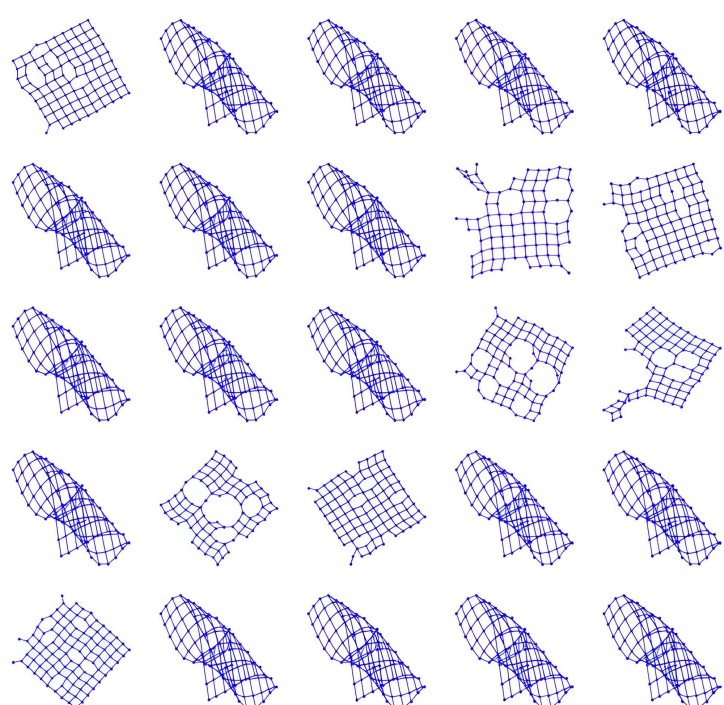

Figure 2: Non-curated structured grid graphs generated by our method, trained with 50 diffusion steps per block. The samples display mostly regular grid-like topology with occasional geometric perturbations, demonstrating the model's ability to capture both structure and variation without any filtering.

### A.11.1 Autoregression in Diffusion Models: Essential or Excess?

Although denoising diffusion models are naturally permutation-invariant, we examine whether incorporating an AR structure offers tangible benefits. Specifically, we investigate whether decomposing the graph generation process into block-wise conditional distributions—based on a structural partial order—can lead to improved quality, efficiency, and stability. To this end, we perform an ablation study by varying the hop radius $K_h$, which defines the granularity of autoregressive blocks. When $K_h = 0$, the graph is treated as a single undivided structure—this corresponds to pure diffusion without any AR decomposition. Larger values of $K_h$ yield finer block-wise partitions, introducing more AR steps. We also evaluate a variant where diffusion is performed without AR but with a larger number of denoising steps, to control for potential improvements from increased sampling. Across all settings, we report molecule validity, uniqueness, atomic and molecular stability, and the FCD. The results, summarized in Table 5, indicate that autoregressive diffusion significantly enhances generation quality. Notably, PARDiff with $K_h = 3$ achieves the best performance with fewer total diffusion steps compared to non-AR setups. This confirms that AR decomposition provides stronger inductive bias, improved stability, and more efficient training—even in permutation-invariant settings.

Table 5: Ablation on QM9 under different autoregressive granularities $K_h$. More blocks (higher $K_h$) improve performance.

| $K_h$ | Steps | Blks | Size | Val. | Uni. | Mol-Stab | Atm-Stab | FCD |
|---|---|---|---|---|---|---|---|---|
| 0 | 140 | 1 | 23.4 | 93.1 | 95.7 | 76.2 | 97.5 | 2.15 |
| 0 | 280 | 1 | 23.4 | 94.0 | 96.2 | 78.1 | 97.8 | 1.84 |
| 0 | 490 | 1 | 23.4 | 94.8 | 96.6 | 78.3 | 98.0 | 1.69 |
| 1 | 140 | 4.1 | 5.7 | 97.3 | 96.7 | 86.8 | 98.4 | 1.21 |
| 2 | 140 | 6.2 | 3.8 | 97.5 | 96.5 | 87.0 | 98.6 | 1.13 |
| 3 | 140 | 8.0 | 3.1 | **97.8** | **96.9** | **88.2** | **98.9** | **0.96** |

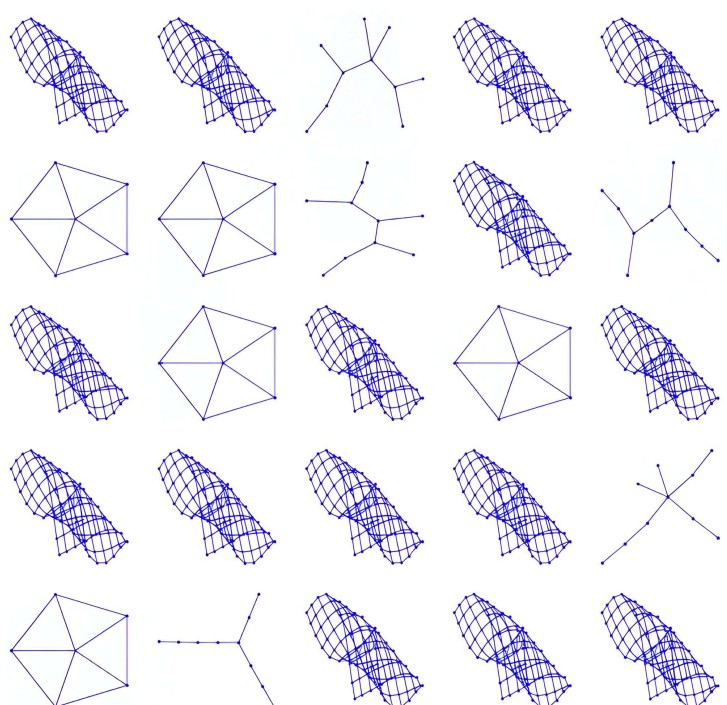

Figure 3: Unfiltered grid-like graphs generated by the eigenvector-enhanced model trained with 50 steps per block.

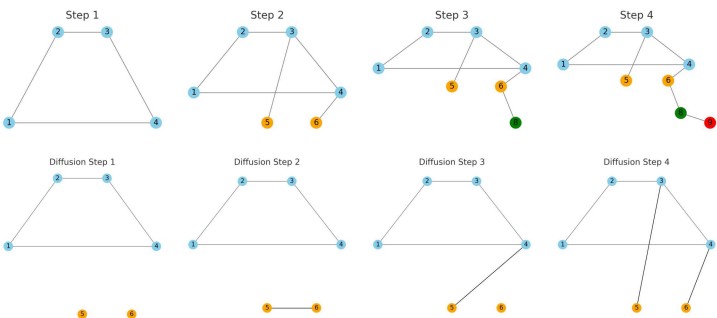

Figure 4: Comparison of autoregressive and diffusion-based graph generation. The top row illustrates autoregressive generation, where nodes and edges are sequentially added in each step. The bottom row shows diffusion-based generation, where the graph is iteratively refined from a noisy initialization toward the target structure.

## A.12 ABLATION STUDY

Table 5 investigates the effect of varying autoregressive granularity, controlled by the number of hierarchical blocks $K_h$, on generation quality in the QM9 dataset. When $K_h = 0$, the model generates the entire graph in a single step, yielding lower performance across all metrics. Increasing the number of diffusion steps improves results incrementally (e.g., FCD drops from 2.15 to 1.69 as steps increase from 140 to 490), but this comes at the cost of significantly higher computational burden, with no structural decomposition. In contrast, introducing even a moderate level of autoregressive structure ($K_h = 1$) immediately boosts performance across all axes—validity, stability, and FCD—indicating that decomposing the graph into substructures introduces useful inductive bias that guides generation more effectively.

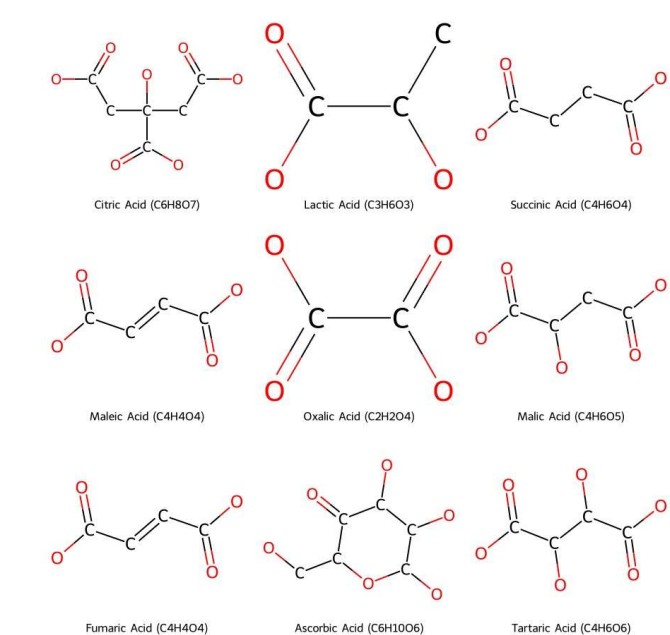

Figure 5: Sample complex molecular structures are generated using PARDIFF.

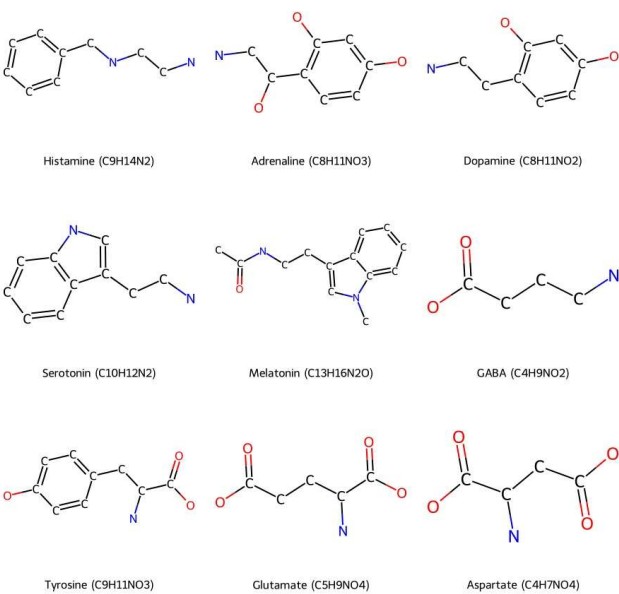

Figure 6: Sample complex molecular structures are generated using PARDIFF.

As $K_h$ increases further, the model progressively refines its granularity of generation, leading to more stable and chemically plausible molecules. At $K_h = 3$, where the graph is generated in 8 blocks, the model achieves its best overall results: highest molecular validity (97.8%), atom stability (98.9%), and the lowest Fréchet ChemNet Distance (0.96). This trend demonstrates that finer-grained autoregressive modeling enables the diffusion process to better condition on intermediate structural context, capturing both local and global dependencies. Figures 8 through 13 show representative samples produced by our model, covering a wide range of organic molecules, including acids, amines, aromatics, and biologically relevant compounds. By progressively adding blocks

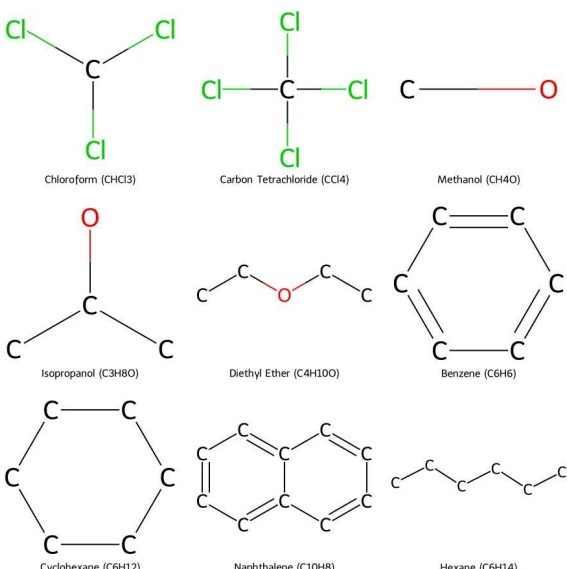

Figure 7: Sample complex molecular structures are generated using PARDIFF.

Figure 8: Sample complex molecular structures are generated using PARDIFF.

of symmetrically ranked nodes and leveraging a permutation-invariant diffusion process, PARDIFF preserves both structural diversity and chemical consistency during generation, capturing intricate bonding patterns with high fidelity.

**Corollary 1.** *Expressivity Bound via WL Test: Since permutation-equivariant neural networks cannot distinguish nodes within the same automorphism orbit of $G$, their discriminative power is upperbounded by the coarsest refinement of these orbits achievable through neighborhood aggregation. In particular, the expressive capacity of message-passing GNNs aligns with the $1$-dimensional Weisfeiler–Lehman (1-WL) test: $u \sim_{WL} v \implies \Phi(u) = \Phi(v)$.*

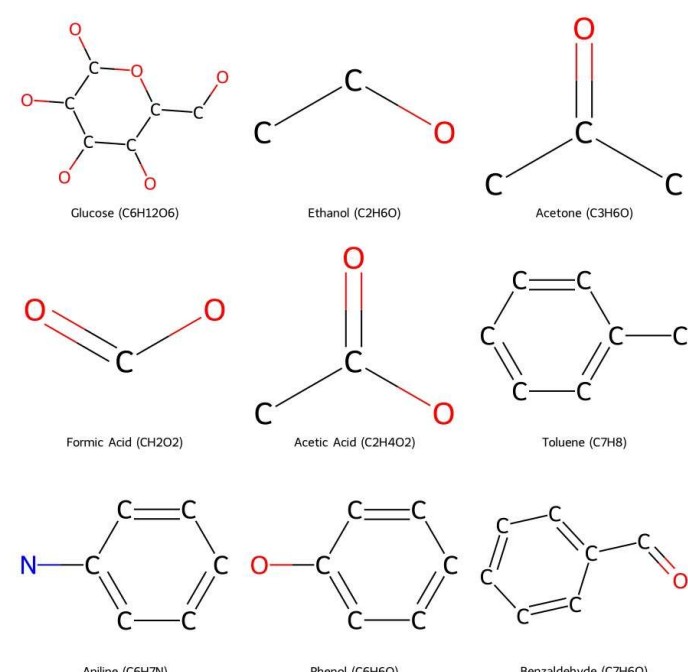

Figure 9: Sample complex molecular structures are generated using PARDIFF.

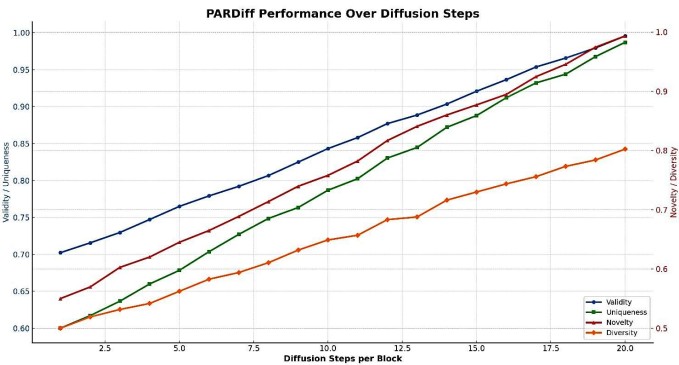

Figure 10: Non-curated structured grid graphs generated by PARDIFF, trained with 50 diffusion steps per block. The samples display mostly regular grid-like topology with occasional geometric perturbations, demonstrating the model's ability to capture both structure and variation without any filtering.

*Proof.* We prove the 1-WL implication by induction over layers and then conclude the orbit claim. Let $w \in \mathcal{V}$ is the raw input features of one specific node of the graph $G = (V, E)$. $\mathbf{x}_w \in \mathbb{R}^d$ is the input feature vector of node $w$. For example, in a molecular graph, $\mathbf{x}_w$ might encode atom type, charge, etc. Let the MPNN have $L$ layers with following updates:

$$\mathbf{h}_w^{(0)} = \psi(\mathbf{x}_w); \ \mathbf{h}_w^{(l+1)} = U(\mathbf{h}_w^{(l)}, A(\mathbf{h}_t^{(l)} : t \in \mathcal{N}(w))), \quad (58)$$

where $A$ is a permutation-invariant multiset aggregator and $U$ a shared update; $\Phi(w) = \mathbf{h}_w^{(L)}$. Let $c_w^{(k)}$ denote the 1-WL color of node $w$ after $k$ rounds:

$$c_w^{(0)} = \text{Hash}(\mathbf{x}_w); \ c_w^{(k+1)} = \text{Hash}(c_w^{(k)}, c_t^{(k)} : t \in \mathcal{N}(w)). \quad (59)$$

Induction hypothesis. Suppose for some $k \leq L$; $c_u^{(k)} = c_v^{(k)} \implies \mathbf{h}_u^{(k)} = \mathbf{h}_u^{(k)}$.

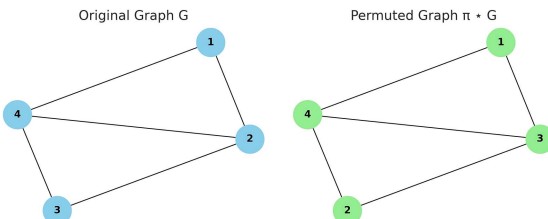

Figure 11: Illustration of permutation-consistency (equivariance, in Theorem 2) in node ranking using a 4 node graph. *Left*: The original graph $G$, where each node is annotated with its ranking value $\psi(u)$. *Right*: The permuted graph $\pi \star G$, obtained by swapping nodes *2* and *3*. The ranking values move consistently with the node labels, showing that $\psi(\pi \star G) = \pi \star \psi(G)$. The structure and relative ordering are preserved under relabeling, demonstrating the permutation-invariance property of Algorithm 1.

1. Base ($k = 0$). If $c_u^{(0)} = c_v^{(0)}$, then $\mathbf{x}_u$ and $\mathbf{x}_v$ are in the same attribute class; since $\psi$ is shared, $\mathbf{h}_u^{(0)} = \psi(\mathbf{x}_u) = \psi(\mathbf{x}_v) = \mathbf{h}_v^{(0)}$.

2. Step ($k \to k + 1$). Assume $c_u^{(k+1)} = c_v^{(k+1)}$. By 1-WL's update, we must have both

$$c_u^{(k)} = c_v^{(k)}; \ \left\{ c_t^{(k)} : t \in \mathcal{N}(u) \right\} = \left\{ c_t^{(k)} : t \in \mathcal{N}(v) \right\} \tag{60}$$

By the induction hypothesis, $c_u^{(k)} = c_v^{(k)}$ implies $\mathbf{h}_u^{(k)} = \mathbf{h}_v^{(k)}$. Moreover, the multiset equality of neighbor colors implies (again by the hypothesis applied elementwise) that the multisets of neighbor embeddings coincide:

$$\left\{ \mathbf{h}_t^{(k)} : t \in \mathcal{N}(u) \right\} = \left\{ \mathbf{h}_t^{(k)} : t \in \mathcal{N}(v) \right\}. \tag{61}$$

Applying the permutation-invariant aggregator $A$ to equal multisets yields equal aggregated messages, and then the shared update $U$ gives:

$$\begin{aligned} \mathbf{h}_u^{(k+1)} &= U\left( \mathbf{h}_u^{(k)}, A\left( \mathbf{h}_t^{(k)} : t \in \mathcal{N}(u) \right) \right) \\ &= U\left( \mathbf{h}_v^{(k)}, A\left( \mathbf{h}_t^{(k)} : t \in \mathcal{N}(v) \right) \right) = \mathbf{h}_v^{(t+1)}. \end{aligned} \tag{62}$$

Thus the claim holds for $k + 1$. By induction, for any $L$, $u \sim_{\text{WL}} v \implies \mathbf{h}_u^{(L)} = \mathbf{h}_v^{(L)}$, i.e., $\Phi(u) = \Phi(v)$.

Automorphism orbits. 1-WL is permutation-invariant; in particular, it assigns equal colors to nodes in the same automorphism orbit (an automorphism maps neighborhoods bijectively at every radius). Hence the 1-WL partition is a coarsening of the orbit partition, and the argument above shows MPNNs cannot refine beyond the WL partition. Therefore permutation-equivariant MPNNs cannot distinguish nodes within the same orbit, nor any pair that 1-WL fails to separate. This proves the corollary. □

Corollary 1 makes explicit that the theoretical ceiling for most GNN architectures is the WL color refinement procedure. (*1*) Automorphism orbits define the hard limit: nodes indistinguishable under symmetry will always collapse to identical embeddings. (*2*) The WL hierarchy shows the algorithmic limit: even when automorphisms are broken, message-passing can at best refine equivalence classes to the 1-WL partition; and (*3*) Consequently, higher-order GNNs (*e.g.*, $K$-WL-GNNs) or symmetry-breaking techniques (*e.g.*, random features, positional encodings, anchor nodes) are required to exceed the expressivity of 1-WL. This bridges group theory (automorphisms), graph theory (orbit partitions), and deep learning (GNN expressivity), offering a unified lens on why standard GNNs fail on hard isomorphism cases such as strongly regular graphs or CAI−FÜRER−IMMERMAN (CFI) graphs Wang et al. (2023) (see Fig. 12) .

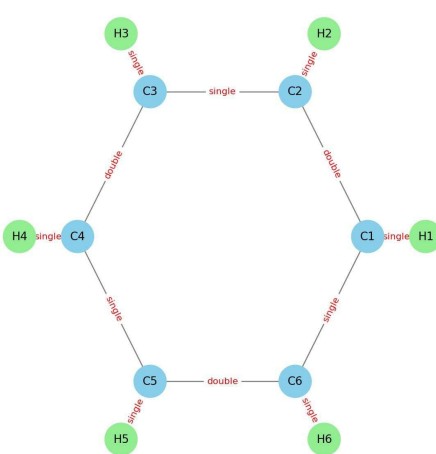

Figure 12: Illustration of orbit equivalence under graph automorphisms. Shown is a 6-cycle graph $C_6H_6$ (Benzene ring), where nodes sharing the same color belong to the same orbit under the automorphism group $\text{Aut}(G)$. Any permutation-equivariant GNN assigns identical embeddings to nodes within the same orbit, regardless of its depth or capacity. This highlights the fundamental expressivity limitation: GNNs cannot distinguish structurally symmetric nodes without additional symmetry-breaking features or higher-order mechanisms (*e.g.*, $K$-WL refinements).