# OpenReview forum: "PARDiff: Bridging Autoregressive and Diffusion Models for Order-Agnostic Graph Generation"
_ICLR.cc/2026/Conference — ICLR 2026 Conference Withdrawn Submission_

### Official Review · Reviewer_DdtL · 2025-11-04

**Soundness:** 1
**Presentation:** 1
**Contribution:** 1
**Rating:** 0
**Confidence:** 5

**Summary:**

LLM generated paper.

**Strengths:**

NO

**Weaknesses:**

Generated wrong stuffs.

**Questions:**

NA

---

### Official Review · Reviewer_wAS7 · 2025-11-04

**Soundness:** 1
**Presentation:** 1
**Contribution:** 1
**Rating:** 0
**Confidence:** 5

**Summary:**

Upon reviewing the manuscript, I discovered a profound degree of content overlap and high similarity with an independently published paper titled "PARD," accessible via arXiv (https://arxiv.org/pdf/2402.03687). This "PARD" paper is accepted by NeurIPS 2024.

Crucially, the current submission fails to cite the "PARD" paper, despite the substantial equivalence in content.

This situation strongly suggests a potential violation of academic ethics, specifically in the form of dual submission or academic plagiarism/self-plagiarism.

I request that you formally investigate this case by reviewing the content of both submissions and their publication status to ascertain whether a violation of the academic code of conduct has occurred.

**Strengths:**

None,

**Weaknesses:**

I have prepared a precise, concise list detailing some of the apparent similarities between the submission (which I'll refer to as "PARDiff") and the "PARD" paper to facilitate your assessment: Please note that this is not an exhaustive list of all potential overlaps, but rather highlights the most evident commonalities.

Introduction Overlap: Lines 50-60 of the PARDiff submission are substantially similar to the last two paragraphs of the Introduction section in the PARD paper.
Methodology/Section 3.1 Similarity: Lines 90-124 of PARDiff exhibit significant overlap with Section 3.1 of the PARD paper.
Theoretical Duplication: Theorem 1 in the PARDiff submission is identical to Proposition 3.1 as presented in the PARD paper.
Architectural/Blockwise Similarity: Section 2.1.1 of PARDiff is substantially similar to the "Autoregressive Blockwise Generation" component within Section 3.2 of the PARD paper.
Section 4 Similarity: Section 2.4 of the PARDiff submission aligns directly with the content of Section 4 of the PARD paper.

**Questions:**

I have prepared a precise, concise list detailing some of the apparent similarities between the submission (which I'll refer to as "PARDiff") and the "PARD" paper to facilitate your assessment: Please note that this is not an exhaustive list of all potential overlaps, but rather highlights the most evident commonalities.

Introduction Overlap: Lines 50-60 of the PARDiff submission are substantially similar to the last two paragraphs of the Introduction section in the PARD paper.
Methodology/Section 3.1 Similarity: Lines 90-124 of PARDiff exhibit significant overlap with Section 3.1 of the PARD paper.
Theoretical Duplication: Theorem 1 in the PARDiff submission is identical to Proposition 3.1 as presented in the PARD paper.
Architectural/Blockwise Similarity: Section 2.1.1 of PARDiff is substantially similar to the "Autoregressive Blockwise Generation" component within Section 3.2 of the PARD paper.
Section 4 Similarity: Section 2.4 of the PARDiff submission aligns directly with the content of Section 4 of the PARD paper.

---

### Official Review · Reviewer_PndM · 2025-11-04

**Soundness:** 2
**Presentation:** 2
**Contribution:** 1
**Rating:** 0
**Confidence:** 4

**Summary:**

Review

The paper appears to be a close derivative of the previous work [1], shows strong signs of being generated by a large language model, and potentially breaks double-blindness by exposing author names. Therefore, I recommend that the paper be rejected and, if the suspicions are confirmed, that the authors face appropriate consequences.

Issue 1: Plagiarism
The submission reproduces core ideas and results from PARD (arXiv:2402.03687v3) while no citation to PARD appears in PARDiff's references. Concretely:
- The block-wise AR factorization and the structural-order ranking via with the same algorithm (PARDiff Sec. 2.1.1, p.3-4; PARD Eq. (4)-(5), p.5).
- The "equivariant-orbit collapse" theorem and the annealing/energy argument for symmetry breaking (PARDiff Sec. 2.2.1-2.3, p.5-6; PARD Sec. 3.3-3.4, p.6-7).
- The causal parallel training and identical masked bilinear product (PARDiff Sec. 2.4, p.7; PARD Sec. 4.2, Eq. (10), p.8; Appx. A.10)
- Matching Algorithms 2-4 for block-size prediction, diffusion training, and generation (PARDiff p.4-6; PARD Appx. A.8 p.18-19)
- The same datasets/splits/metrics with reproduced baseline rows in Tables 1-3 (PARDiff p.8-9; PARD p.9-10).

Issue 2: LLM generation
There are some strong indicators that the paper may have been generated using a large language model:

- The paper cites a non-existent reference for ConGress, which was introduced as part of [2].  The URL is invalid and there is no such publication in the ICRL 2023 proceedings. The authors refer to it as:
	- "Chen Cai and Yusu Wang. Congress: Conditional graph generation via score-based diffusion. In International Conference on Learning Representations (ICLR), 2023. URL https:// [openreview.net/forum?id=ycyWpR0Uxn](http://openreview.net/forum?id=ycyWpR0Uxn)."

- DiGress is cited twice, both times referring to the same arxiv link. The second version adds M. Bronstein, who is not an author:
	- Clement Vignac, Igor Krawczuk, Antoine Siraudin, Bohan Wang, Volkan Cevher, and Pascal Frossard. Digress: Discrete denoising diffusion for graph generation. arXiv preprint arXiv:2209.14734, 2022a.
    - Clement Vignac, Jiaxuan You, Jure Leskovec, and Michael M. Bronstein. Digress: Discrete denoising diffusion for graph generation. In International Conference on Learning Representations (ICLR), 2023. URL [https://arxiv.org/abs/2209.14734](https://arxiv.org/abs/2209.14734).

- In Table 1. with QM9 results the row titled "Dataset (Optimal)" is strange. The reported "optimal" values are exceeded by the proposed method, which is logically inconsistent. The "Mol" column ("molecular accuracy" - not explained what it means) assigns the dataset itself a score of 87%, implying the dataset "predicts itself," which is meaningless and unexplained.

Issue 3: violation of double-blindness
There is a name of the author of [1] in the LICENSE file of the submission. While not definitive, the name could correspond to an author of this submission. It is also possible that this is an artifact of copying the LICENSE or other metadata files from the original PARD repository, which would itself further confirm plagiarism.

References:
[1] - Zhao, L., Ding, X., & Akoglu, L. (2024). PARD: Permutation-invariant Autoregressive Diffusion for Graph Generation. arXiv:2402.03687v3.
[2] - Vignac, C., Krawczuk, I., Siraudin, A., Wang, B., Cevher, V., & Frossard, P. (2023). DiGress: Discrete denoising diffusion for graph generation. In International Conference on Learning Representations (ICLR 2023)

**Strengths:**

.

**Weaknesses:**

.

**Questions:**

.

---

### Official Review · Reviewer_W42x · 2025-11-08

**Soundness:** 2
**Presentation:** 2
**Contribution:** 2
**Rating:** 2
**Confidence:** 4

**Summary:**

This paper proposes PARDiff, a progressive autoregressive–diffusion framework for graph generation. The key idea is to bridge the controllability of autoregressive (AR) models and the permutation-invariance of diffusion models by dynamically decomposing graphs into topological blocks, predicting the block order autoregressively, and generating each block via a shared equivariant discrete diffusion process. This allows the model to generate large graphs in a scalable, order-informed manner while avoiding permutation bias. Experiments on molecular and synthetic graph datasets demonstrate improved controllability and structural coherence compared to prior AR-only, diffusion-only, and hybrid approaches.

**Strengths:**

The ability to improve scalability, structure-awareness, and expressivity in graph generative models is valuable for molecular design and structured graph synthesis tasks.

**Weaknesses:**

The block decomposition module plays a central role but is not fully interpretable. The model may produce suboptimal or unstable block partitions, which directly affect generation quality.

The training process involves multiple coupled learning components (block prediction, autoregressive ordering, diffusion generation), which increases implementation complexity and may hinder reproducibility.

Although more efficient than full-node autoregression, inference still requires step-wise block generation, which can be slow for very large graphs.

The paper mainly evaluates datasets where block-like or community structures are naturally present. It is unclear how well the model performs on graphs that lack such modularity.

**Questions:**

NA

**Details Of Ethics Concerns:**

It is indicated by other reviewers, this paper seems generated by the LLM.

---

### Note · Authors · 2025-11-20

I have read and agree with the venue's withdrawal policy on behalf of myself and my co-authors.